# Fern Spores—“Ready-to-Use” Standards for Plant Genome Size Estimation Using a Flow Cytometric Approach

**DOI:** 10.3390/plants12010140

**Published:** 2022-12-27

**Authors:** Sheng-Kai Tang, Pei-Hsuan Lee, Wei-Ting Liou, Chen-Hsiang Lin, Yao-Moan Huang, Li-Yaung Kuo

**Affiliations:** 1Institute of Molecular and Cellular Biology, National Tsing Hua University, Hsinchu City 300, Taiwan; 2Taiwan Forestry Research Institute, 53 Nan-Hai Road, Taipei City 100, Taiwan; 3Experimental Forest, College of Bio-Resources and Agriculture, National Taiwan University, Nantou County 557, Taiwan; 4Department of Life Sciences, National Chung Hsing University, Taichung City 402, Taiwan; 5Taoyuan District Agricultural Research and Extension Station, Council of Agriculture, Executive Yuan, Taoyuan City 327, Taiwan

**Keywords:** bead vortex, C-values, fern, flow cytometry, genome size, spores

## Abstract

Spores and pollen of plants were used as flow cytometric materials to efficiently infer genome sizes. Given this advantage, they hold great potential for various flow cytometric applications, particularly as plant genome size standards. To develop such novel standards, we investigated conditions of pretreatment (bead vortex), buffer, and reliable genome sizes of three fern spore collections—*Cibotium taiwanense* “*Kuo4395*”, *Sphaeropteris lepifera* “*Tang0001*”, and *Alsophila metteniana* “*Lee s.n.*”. Additionally, up to 30 year-old spore collections were obtained from herbarium specimens and from samples stored at 4 °C; their spore nuclei were extracted, and the quality and quantity of these nucleus extractions through storage ages were examined. Nuclear extractions with a longer bead vortex duration or lower spore/bead ratio generally resulted in a higher recovered quantity but a lower quality or purity. For each spore standard, the protocol optimization was determined by their performance in bead vortex conditions, and a 1C genome size was further inferred by linear regression (*C. taiwanense* “*Kuo4395*” = 5.058 pg; *S. lepifera* “*Tang0001*” = 7.117 pg; and *A. metteniana* “*Lee s.n.*” = 19.379 pg). Spore nucleus quality and quantity are significantly negatively correlated with storage ages. Nuclear extractions of 10-year-old refrigerated spores remained qualified as a genome size standard; however, none of the herbarium spore collections fit such criteria. Our study is the first to develop and apply dried and refrigerated spores for genome size standards. These standards are ready to use, easy to manipulate, and feature long-term storage in comparison with traditionally used standards of fresh leaves.

## 1. Introduction

With the advances of genome sequencing technologies, exploration of genomic diversity outside of model organisms is no longer restricted. Meanwhile, prior knowledge of genome size across the tree of life became increasingly important. Genome size information is critical for biologists to select candidate species or strains that are cost-effective for whole-genome sequencing. Additionally, this information provides a basis for further works to study genomic organization, regulation, and evolution [1,2]. This is particularly true for plants because these organisms evolved varied genome size, over 2000-fold, and were marked as the largest among eukaryotes [3,4,5]. The significant variation in genome size among land plants is predominantly driven by whole-genome doubling, thus revealing rounds of polyploidization in their evolutionary histories [6]. Compared to diploids, polyploids, due to their enlarged genome size, might cost double (or more) to complete whole genome sequencing, and are thereby undesirable for such projects. By far, flow cytometry (FCM) proves the most efficient and cost-effective approach to infer accurate genome size in plants [7,8]. Such flow cytometric investigation requires extractions of nuclei from standard(s) with a known genome size (or C-value). Plant standards of this purpose started being developed in the 1980s [9], and several well-established C-value standards belong to certain named strains of crops (e.g., [10,11,12]). In addition to the features of being easy and fast growing, genomic contents are believed to be almost identical from generation to generation in these crop strains. However, the maintenance of “living” standards is necessary in order to retrieve fresh leaves from the same plants for nuclear extraction.

Within the past few decades, FCM methods for plant C-value estimates were not only applied to fresh leaf tissue, but also succeeded in other types of organs, such as seeds [13,14,15], pollen [16,17,18,19], and spores [20,21]. The FCM experiments for pollen and spores were demonstrated to be more efficient than using fresh leaves [18,20]. For example, the bead vortex approach by Kuo and Huang [22] can complete nuclear extractions of a dozen or more spore samples within 2 min, whereas it usually takes more time to chop tissue from a leaf sample. Such methodology of bead vortex continuously succeeded, especially in fern spores [21,23,24,25]. Another great advantage of such “dried” fern spore material is that neither fresh tissue nor a living plant is required for C-value estimation [22]. Moreover, a very small amount of spores is usually required for a FCM experiment, and such an amount can be easily harvested from most fern species [26]. For example, hundreds of millions of spores (>10 g) can be collected from a single leaf of a fern tree [27]. For a single FCM reaction, only 7 mg of spores is required [22], thus a spore collection from a single fern leaf can support thousands of FCM reactions. Last, selfing lines can be generated in most ferns [28,29,30], and are therefore prone to be genetically conserved across generations. This advantage raises the repeatability of genome size estimates. In sum, because of being methodologically efficient, abundantly harvested, dry-stored, and genetically stable, spores of these plants hold a great potential to be developed as a new kind of “ready-to-use” C-value standard.

In addition to the above-mentioned considerations, requirements such as range and precision of genome size estimates should be taken into account when developing an ideal C-value standard [7]. First, an applied standard must have a genome size close enough to that of the target species in order to reduce the risk of estimate error [12,31,32]. Second, chemical interaction coming from buffers and tissue extraction is prone to affect the degree of DNA staining within nuclei, and to have subsequent influence on further estimation of genome size [33]. In addition, some other factors should be considered, particularly for a fern/lycophyte “spore” standard. Kuo et al. [34], for instance, adjusted vortex duration, spore amount, and bead amount for bead vortex pretreatment in order to optimize conditions for nuclear extraction. Last, storage age is one of the most important factors affecting spore viability, and, likewise, quality of spore nuclei. Spore viability in some ferns is known to be strikingly decreasing through time [35,36], while it is still unknown whether spore nuclear quality declines in a similar manner.

To develop spores as new C-value standards for plants, we chose ferns for our study because a high quantity of spores per individual is easily accessible in these plants [26]. We first examined spore traits, including their diameter, mass, nucleus size, and genome size in different fern species, and selected three collections—*Cibotium taiwanense* C.M.Kuo “*Kuo4395*”, *Sphaeropteris lepifera* R.M.Tryon “*Tang0001*”, and *Alsophila metteniana* Hance “*Lee s.n.*” (Table 1) as candidates for the downstream assessment (using the species epithets as their names in the following text). Further, in order to optimize extraction protocol of their spore nuclei, we tested different bead vortex conditions, and determined the best acquisition with either a considerable quality or quantity for spore nuclei among them. Then, to obtain a reliable C-value for future application, we designed several sets of buffers and internal standards. Last, to check integrity of spore nuclei through storage age and conditions, we used additional spore collections of *S. lepifera,* which were either stored at 4 °C for up to 20 years, or in a herbarium at room temperature for up to 30 years. With the above experiments, three promising and easy-to-use spore standards of C-value estimation are presented in this study.

## 2. Results

### 2.1. Spore Traits and FCM Performances

The spore traits are summarized in Table 1. The spores of all the three species are similar in shape and their equatorial diameters. The nucleus sizes positively correlate with their genome sizes, and both change in the same order of *A. metteniana* > *S. lepifera* > *C. taiwanense* (Table 1); however, the changes of spore size (i.e., equatorial diameter) and mass are shown with a different trend that does not correspond to their nucleus or genome size. Specifically, their intracellular space occupied by a nucleus tends to be different in proportion. For example, spores of *C. taiwanense* have the smallest nuclei among the three while both its spore sizes are the largest. As a result, a spore nucleus occupies a relatively small space within a *C. taiwanense* spore compared to that in the other two species. 

Generally, most trends for the resulting quality and quantity match with our expectations (Figure 1 and Figure 2; Appendix A), and vortex duration is an important contribution to bead vortex results (Table 2). The results are similar between *S. lepifera* and *A. metteniana*, in which longer vortex duration was usually accompanied by higher coefficient variation (CV), non-nuclear particles (*np*), and relative recovery rate (*rrr*) values (Figure 2 and Appendix A; see “*4.2. Protocol optimization*” for the calculation of these values), several of which are also significantly supported by the ANOVA results (Table 2). However, the *rrr* value decreased during the higher spore/bead ratio, and showed a clear opposite trend against our expectation (Figure 2). This pattern instead implies that an increase in spore amount in a tube could not proportionally increase the quantity of nucleus particles. The peak of the *rrr* value might appear in a low spore–bead ratio, which is likely beyond the ratio values surveyed in the current study. Last, patterns in *C. taiwanese* suggest that none of the treatments can prevent a high *np* result under bead vortexing (Figure 3). The pattern coordinates with its low ratio of spore nuclear size and spore size, and implies that properties of infra-sporic structures in certain species might affect the quality of nuclear extraction.

Based on these performances, shorter vortex duration and higher spore/bead ratio is apparently best for yielding high quality, while longer vortex duration and lower spore/bead ratio is best for yielding high quantity (i.e., high extraction efficiency). In *C. taiwanese*, given that all treatments consistently resulted in high *np* values and similar CV values, access to a high relative recovery rate becomes the most important consideration. Regarding this, a 2 min-vortex with a 3.5 mg-spore and 20 beads (the set of 2, 3.5, and 20) is recommended because it is shown to receive the highest relative recovery rate (Appendix A). In *S. lepifera*, both great quality and quantity can be maintained under the set of (1, 3.5, and 20), which is shown to receive a high relative recovery rate and a moderate proportion of non-nuclear particles (Appendix A). In *A. metteniana*, the condition of (1, 3.5, and 16) performed moderately in both quality and quantity of nuclear extraction, where the *rrr* was high while the *np* was low (Appendix A).

### 2.2. Genome Size Estimate

Intriguingly, C-value estimates of *A. metteniana* and *C. taiwanese* deviate between spores and leaves (Table 3 and Table 4). These deviations were likely caused by tissue-specific effects (see the detailed explanation in Discussion). Nonetheless, with the use of spore materials, C-value estimates using different buffers and internal standards are mostly consistent, except when using *Isoetes taiwanensis* De Vol for *S. lepifera* (Table 3). Aside from this extreme value, the estimates inferred by linear regression (Table 5) are median among all estimates, and show less than 5% deviation from the estimates with marginal ones (Table 3). Regarding the CV values, <5% were found among three spore collections, which implies the universality of buffer application for spore FCM application. However, buffers seemed to perform slight differences among spore collections (Table 3). For instance, the LB01 buffer resulted in the lowest CV values in *C. taiwanense* and *S. lepifera*, but highest in *A. metteniana*, which implies that buffer performance of genome size estimates is also affected by an interaction between buffer and species. In the future, we recommend the use of 1C-values by linear regression (Table 5) as the main genome size references for these spore standards, and referencing the buffer results (Table 3) if any slight difference of buffer performance is relevant.

### 2.3. Effects of Storage Age

All spore collections from herbarium specimens failed to detect an unambiguous fluorescence signal of nuclei, so only the 4 °C-stored samples were included in the following correlation analyses (Figure 3). We found that older spores resulted in significantly higher CV values and lower relative recovery rate. Non-nuclear particles are positively, but not significantly, correlated with storage ages. It is suggested that 4 °C storage age should not exceed ten years in order to get an acceptable CV value (<5%) and relative recovery rate. The oldest sample satisfied with these criteria has a storage age of 13 years. Although it resulted in a CV value of 4.28%, nuclear extraction of a single extraction was sufficient for the requirement of the FCM standard.

## 3. Discussion

### 3.1. How Does Fern Spore Standard Perform Better?

Our study demonstrates how fern spore standards perform well and even outperform plant leaf standards in several aspects. First, the fast and easy-to-use bead vortex method for fern spore materials is more efficient than chopping methods for leaf tissue. This feature additionally allows operators to minimize bias during destruction of the materials. Second, in terms of accessibility and availability, sufficient intact nuclei can be easily extracted from long-term preserved materials (Figure 3). In our tests, even the fern spores which were dry-stored under 4 °C for more than ten years were still workable, and the resulting quality and quantity of these nucleus extractions satisfied criteria for C-value estimation. This allows fern spores to serve as ready-to-use and easy-to-maintain standards. In comparison, breeding and maintenance of living plants required for leaf standards is labor intensive, and also requires certain environmental conditions and a specific space for growing plants. Importantly, most of these previously established standards are annual crops favoring the temperate climate. Sustainable accesses to fresh leaves from these plants, therefore, need a well-controlled environment to cultivate them for generations. Once lacking a temperature-controlled growing chamber or a green house, it is difficult for a non-temperate place to maintain such breeding conditions, particularly for their flowering and fruiting. Third, these fern spores feature stable DNA stain results by crossing mixtures with various leaf tissue extractions (i.e., the internal standards here) and various buffers (Table 3). In addition, the genome size estimate seems more accurate when using fern spores instead of conspecific leaf tissue. In the case of *C. taiwanese*, the estimates from genome sequencing results (L.-Y. Kuo pers. comm.) are much closer to the sporic 1C-value than the leaves. As for leaf materials, previous studies suggest that 1C-value estimates using leaf materials are usually lower than those using spores, and seemingly underestimated [18,20,21,23,24]. Deviations of genome size estimates were revealed from leaves with different tissue conditions [37,38,39], and are presumably caused by different cell stages and different levels of chromatin condensation within nuclei [39]. In particular, chromatin is found increasingly decondensed in plant nuclei during the transition from a vegetative fate into a reproductive one; for example, sporogenesis of spore mother cells [40], and thus become more accessible to DNA staining chemicals.

### 3.2. Extending Candidates of Spore Standards, and Protocol Optimization

Ideally, the genome size range of emerging spore standards is able to cover the range of land plants. The three new spore standards developed here have 1C-values between 5 and 20 pg. In comparison, common leaf standards’ are between 0.15 and 15 pg [31,41]. Spore-bearing vascular plants with a smaller genome size (1C DNA content < 2 pg), such as Lycopodiaceae, Selaginellaceae, and Gleicheniaceae [25,42,43], and ones with a larger size, such as Plagiogyriaceae, Pteridaceae, or Ophioglossaceae, whose 1C DNA content are about 20 to 40 pg [21,25,44], are promising standards in the future in order to fill in the current gaps in genome size range. When it comes to accessibility and available quantities, species that can be easily cultivated or have a high spore yield are prioritized over heterosporous species or those with low spore yields, such as Selaginellaceae [26]. When considering the possibility that genome sizes can vary moderately among different individuals or even generations within a species [38,45,46,47,48,49] (Appendix A), initially identifying a single selfed or clone-able strain/individual is critical. Such selection can greatly benefit the reliability of C-value estimation. For instance, the *Dryopteris varia* Kuntze collection mentioned in Materials and Methods is apomictic and fits this perspective well.

Spore traits could also be important factors, and are particularly sensitive to bead vortex protocols. For instance, green spores such as *Osmunda japonica* Thunb., whose nuclei are less protected from physical damage by thin spore perine, are fragile under bead vortexing [20]. Among the cases demonstrated here, a *C. taiwanense* spore nucleus occupies a relatively small space within a spore, which is very different from spores of the other two species (Table 1). This sporic property is likely associated with a high proportion of non-nuclear particles (Figure 2). As a result, when seeking and optimizing protocols for future candidates, we suggest also considering their infra-sporic structure as well as other uninvestigated traits (e.g., spore shape difference, trilete vs. monolete).

### 3.3. Storage Method of Spore for FCM

In this study, spore collections of *S. lepifera* stored for over ten years are available for estimating its genome size. In contrast, regardless of age (8–30 years old), all the herbarium spores of *S. lepifera* used in this study failed to be extracted with sufficient nuclei, and there was no detection of an obvious signal peak corresponding to spore nuclei. These results imply that the storage method of the spore is critical for the success of FCM. Nonetheless, Roberts [26] successfully received clear, blunt fluorescent signals of *Rosa canina* L. pollen and leaf tissue collected from herbarium specimens. Though these materials were relatively fresh (24 months old), they indicated that nuclei of some plants or organs can bear harsh treatments, and will not be completely destroyed shortly after specimen vouchering. In addition to age and (pre)treatments for herbarium specimens, taxon-specific properties seem to be the major factors for FCM results (see below). For instance, eusporangiate structures in some ferns appear to better protect their spores as well as the spore nuclei inside. Even being stored in a herbarium over 20 years, intact spore nuclei can be easily accessed from specimens of such fern taxa [21].

For fern spores, though it is difficult to examine the tissue condition directly, spore viability can be a key indicator—a higher germination rate means a higher proportion of intact nuclei. Previous studies suggest that spore storage conditions can be taxon-dependent in herbaria. For instance, some taxa that contain a sporocarp or live in fire-prone environments have a spore viability of a month to nearly 100 years [50,51,52]. Under dry storage, spore germination rate declines through age, e.g., [53,54], and viability can be better maintained at the lower temperatures [35,55,56,57]. In consequence, we suggest that fern spores used as FCM materials should be refrigerated for long-term preservation. Although a few plant species or their specific organs might be vulnerable to drying procedures and maintenance methods among different herbaria [53], keeping refrigerated duplicates for FCM experiments is highly recommended prior to preparation of herbarium specimens.

## 4. Materials and Methods

### 4.1. Spore Sample Source

In order to develop broadly applicable standards, we reviewed previously reported fern genome sizes [8,25,42], and selected candidates after preliminary tests by quickly checking their flow cytometric performance. Four species were chosen: *Cibotium taiwanense* (*1n* nuclei = 4.51 pg) [25], *Sphaeropteris lepifera* (*1n* nuclei = 6.45 pg) [42], *Alsophila metteniana* [*1n* nuclei = 17.00 pg (this study)], and *Dryopteris varia* (*1n* nuclei = 24~26 pg) [22,58]. However, because the triploid apomictic individual of *D. varia* yielded insufficient spores, we were unable to complete a full assessment for this species. Three spore collections for the three remaining candidate species are detailed in Table 1. These three collections were revealed with the genome sizes (Figure 1) matching that from previous conspecific or congeneric reports with ploidy inferences [42,43]. As a result, *C. taiwanense* “*Kuo4395*”, *S. lepifera* “*Tang0001*”, and *A. metteniana* “*Lee s.n.*” are assumed to be, respectively, a diploid, diploid, and tetraploid. These ploidies are also identical with previous cytological reports of the same species known from the same source region (i.e., Taiwan) [59]. These spore collections were stored at 4 °C less than 6 months before any further experiments. Currently, living plants and spore sources of the three collections are maintained at the Dr. Cecilia Koo Botanic Conservation Center (KBCC), Taiwan Forestry Research Institute (TFRI), and National Tsing Hua University (NTHU). These spore sources can be shared under legal process for academic use, and requested from the corresponding authors (L.-Y.K. and Y-.M.H.) and the Dr. Cecilia Koo Botanic Conservation Center (the fern collection manager, Chun-Ming Chen; email: forestaray@gmail.com). To examine spore nuclear quality through storage age (see below in “*Effects of storage age*”), additional spore collections of *S. lepifera* were used. These samples included those also stored in a 4 °C fridge within a dark room, but with different ages up to 20 years (Appendix A), and that from 8-to-30-year-old specimens stored at the TAIF herbarium under room temperature (air conditioned ~20 °C) (Appendix A). Methods of spore collection and isolation were based on Huang et al. [60].

Spore size, mass, and nucleus size were examined. More than 60 spores for each collection were photographed under a light microscope (WILD M8; Leica, Wetzlar, Germany), and their equatorial diameter was measured using Image Pro plus 5.0.0 (Media Cybernetics, Silver Spring, MD, USA). To measure spore mean mass (*M_X_*), we first weighted spores at *quantum satis* (*M_T_*) using a sensitive microbalance (MX5; Mettler Toledo, Columbus, OH, USA), and diluted them with a fixed magnification (*C_T_*) of distilled water; then counted the concentration of spores (*N_X_*) under a microscope with 10 replicates for inferring *Mx*. To mix each spore solution (in a tube) well, 5-second vortexing was applied before counting.
$$M_{X}=\frac{M_{T}}{N_{X}\times C_{T}}$$

For spore nucleus size, we used 2.3 mm stainless steel beads (BioSpec, Bartlesville, OK, USA) to break the spores for their nuclear extraction. The spore nuclei were dyed with acetocarmine for 10 to 30 min, and then photographed under a light microscope. For each species, 30 spore nuclei were measured by Image Pro plus 5.0.0 (Media Cybernetics, Silver Spring, MD, USA).

### 4.2. Protocol Optimization

Our protocol optimization for spore nuclear extraction mainly followed the designs of Kuo et al. [20], with some further modifications. In total, 18 bead vortex treatments were applied with three spore amounts (3.5, 7, and 10.5 mg per tube), three bead amounts (12, 16, 20 beads *ea*.; 2.3 mm stainless steel beads; BioSpec, Bartlesville, MD, USA), and two vortex durations (1 or 2 min). Three replicates were performed for each bead vortex treatment. Through these treatments, we sought the condition(s) most effective and productive for spore nuclear extraction. The spores and beads were added with 0.25 mL GPB buffer [34] [0.5 mM spermine, 30 mM sodium citrate, 20 mM MOPS, 80 mM KCl, 20 mM NaCl, 0.5% (*v*/*v*) Triton X-100, pH = 7.0; with 0.5% (*v*/*v*) 2-mercaptoethanol, 40 mg mL^−1^ PVP-40, and 0.1 mg mL^−1^ RNaseA added before using] in a 1.5 mL microcentrifuge tube (Gunster Biotech, Taipei, Taiwan). The GPB buffer was chosen for this initial test because it is was known to be widely applicable across fern taxa [25,42,43]. The GPB buffer mixtures were then vortexed at 1900 rpm for either 1 or 2 min. After vortexing, extractions from three different spore collections were filtered through different sieve sizes depending on their spore traits (i.e., spore size and spore nuclear size; Table 1) in order to efficiently remove non-nuclear particles, including intact spores, while allowing spore nuclei to pass through, 20 µm circular nylon filters (Sysmex Partec, Kobe, Hyogo, Japan) were chosen for the extractions of *A. metteniana* and *S. lepifera*, while 10 µm circular nylon filters (Sysmex Partec, Hyogo, Japan) were used for those of *C. taiwanense*. Then, each filtered extraction was divided into two aliquots: one was mixed with the nuclear extraction of internal standard [(*Nicotiana tabacum* ‘Xanthi’; 2C = 10.04 pg] [12] under a fixed volume ratio, while another was not. All of the solutions were finally dyed with a 1:50 volume of propidium iodide (PI) solution (2.04 mg mL^−1^). After incubating at 4 °C in the dark for 1 h, all samples were run on a BD FACSCan system (BD Biosciences, Franklin Lakes, NJ, USA). For each treatment, over 1300 particles were collected for both spore and *2n* nuclei of the internal standard, and all measured peaks of signal had coefficient variation (CV) values less than 5%.

In addition to the CV values of peaks of spore nuclei, we also applied two additional indicators, “non-nuclear particles (*np*)” and “relative recovery rate (*rrr*)”, respectively, to infer purity and quantity of spore nuclear extractions. A lower CV value represents higher quality of nuclei, while greater *np* implies a higher proportion of non-nuclear particles, which likely resulted from over-disruption during bead vortexing and caused noisy signals for flow cytometric inference. For *np* values, the aliquots without internal standard were used for inference. Relative recovery rate (*rrr*) stands for the number of spore nuclei in proportion to its initial input of spore (mg). In other words, a higher *rrr* value means a higher efficiency of extracting spore nuclei per unit of spores. For *rrr* values, the aliquots mixed with internal standard were used. Because the nuclear extraction of the internal standard was prepared from the same stock as for each spore collection, the *2n* nuclear amount of the internal standard should be constant across the same experimental batch.
$$np=1-\frac{spore\;nuclei}{total\;events};$$
$$rrr=\frac{spore\;nuclei}{internal\;standard\;\;2n\;nuclei\times spore\;amount}$$

Together with the CV values, significance of these indicators among different bead vortex treatments was examined using three-way ANOVA and Tukey’s honest significant difference (HSD) test in JMP13 software (SAS Institute Inc., Cary, NC, USA). We expect that longer vortex duration will contribute to higher CV, *np* and *rrr* values. We also expect that the higher spore/bead ratio will lead to higher *rrr* values and lower CV and *np* values ―that is, with consistent bead amounts, the more spores we use, the more nuclei we can acquire. The ideal treatments, with either high quantity (i.e., extraction efficiency), quality, or purity for nucleus acquisition, were decided for each spore collection based on the above statistics.

### 4.3. Genome Size Estimates

To reveal a reliable range of C-value in each of the three spore collections, different buffers and internal standards were used for genome size estimation. The buffers included GPB [34], Beckman [61] [50 mM Na_2_SO_3_, 50 mM Tris-HCl, 1.0% (*v*/*v*) Triton X-100, pH = 7.5], and LB01 buffer [32] [0.5 mM spermine, 80 mM KCl, 20 mM NaCl, 2 mM Na_2_EDTA, 15 mM Tris-HCl, 0.1% (*v*/*v*) Triton X-100, pH = 7.5]. Additionally, 0.5% (*v*/*v*) 2-mercaptoethanol, 40 mg mL^−1^ PVP-40, and 0.1 mg mL^−1^ RNaseA were added to these buffers before use. The conditions of bead vortexing and applied filters for spore nuclei of each species were based on suggestions in “*Protocol optimization*” (see in the previous section). In addition, genome sizes were estimated using the leaf nuclei from the same parental plants, but only performed under the GPB buffer. For the chopping method of leaf materials, we followed Kuo and Huang [22]. Five internal standards were chosen for genome size estimation of spore or leaf nuclei: *Vicia faba* L. cv. Inovec (2C = 26.90 pg), *Secale cereale* L. cv. Dankovske (2C = 16.19 pg), *Nicotiana tabacum* L. cv. Xanthi (2C = 10.04 pg), *Pisum sativum* L. cv. Ctirad (2C = 9.09 pg), and *Isoetes taiwanensis* (2C = 3.368 pg) [12,32,62]. Six replicates were performed for each buffer/standard combination in each collection, and the particle number per peak was set to over 10,000. Other flow cytometric conditions were the same as described (see in the previous section “*Protocol optimization*”). The lowest CV values were reported for buffer selection in each species.

We used two approaches to infer C-value: (1) simply using ratio of fluorescence intensity (i.e., the particles’ mean FL-area) to multiply the known genome size of the internal standard [31,32], and (2) a linear regression with results of fluorescence intensities coming from three different internal standards [31]. The second approach relied on the known genome sizes of the internal standards and their ratios of fluorescence intensities to that of a certain sample. We applied these plots to infer a linear regression line and its formula for each candidate. The resulting formula returned the sample’s genome size value with the ratio of fluorescence intensities set to 1. For this approach, we used only the data coming from the same buffer, and only the GPB buffer was applied.

Notably, because the nuclei of internal standards from leaf tissue and that of our spore samples isolated by different destructive methods (i.e., chopping for leaves and bead-vortexing for spores), these nucleus extractions had to be prepared separately. Standardization from such sample preparation is defined as a pseudo-internal one [31], despite different nucleus extractions that are first mixed to create a homogeneous condition before staining. However, no evidence so far was revealed that the staining property of the nuclei per se can be affected by the different pretreatments [21]. Our preliminary test showed that the estimates, whether the spore nuclei of two samples were extracted together in the same tube or not, are not significantly different. As a result, different processes of nucleus extractions and sample pretreatments were unlikely concerns for the standardization, as well as for the precision of genome size estimation.

### 4.4. Effects of Storage Age

In this test, we used only *S. lepifera* spore collections. The condition of bead vortexing and applied filter were based on suggestions in “*Protocol optimization*” (see in the previous section). LB01 buffer [with 0.5% (*v*/*v*) 2-mercaptoethanol, 40 mg mL^−1^ PVP-40, and 0.1 mg mL^−1^ RNaseA added] was used because in comparison with the other two, this buffer resulted in the lowest CV value for *S. lepifera* spore nuclei, as revealed by the results from “*Genome size estimation*”. *Nicotiana tabacum* cv. Xanthi was used as the internal standard, and added into each sample (same as in “*Protocol optimization*”). Other flow cytometric conditions or experimental details were the same as in “*Protocol optimization*”. CV values, non-nuclear particles (*np*), and relative recovery rate (*rrr*) (see the equations in “*Protocol optimization*”) were applied as indicators of spore nuclear quality. Linear regression against storage age was performed individually for these three parameters using JMP13 software (SAS Institute Inc., Cary, NC, USA).

## 5. Conclusions

This study is the first to develop fern spores at genome size standards for a flow cytometric purpose. The convenience of these new spore standards against traditional leaf ones are emphasized here. To encourage use of these standards, we also provided the protocols and guidelines for them (detailed in *Optimization and suggestion of bead vortex condition*), including their genome size estimate (detailed in *Genome size estimate*). We are looking forward to more fern or even lycophyte spore standards developed in the near future in order to broaden the applicability of this emerging method. Importantly, we hope this convenient tool can also facilitate investigation of C-value diversity in plants that is insightful for plant genomic research.

## Figures and Tables

**Figure 1 plants-12-00140-f001:**
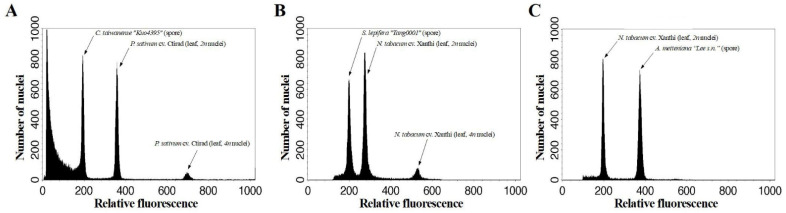
Estimation of (**A**) *Cibotium taiwanense* “*Kuo4395*”, (**B**) *Sphaeropteris lepifera* “*Tang0001*”, and (**C**) *Alsophila metteniana* “*Lee s.n.*” spore genome sizes by flow cytometry using the internal standards of leaf nuclei of *Pisum sativum* cv. Ctirad and *Nicotiana tabacum* cv. Xanthi.

**Figure 2 plants-12-00140-f002:**
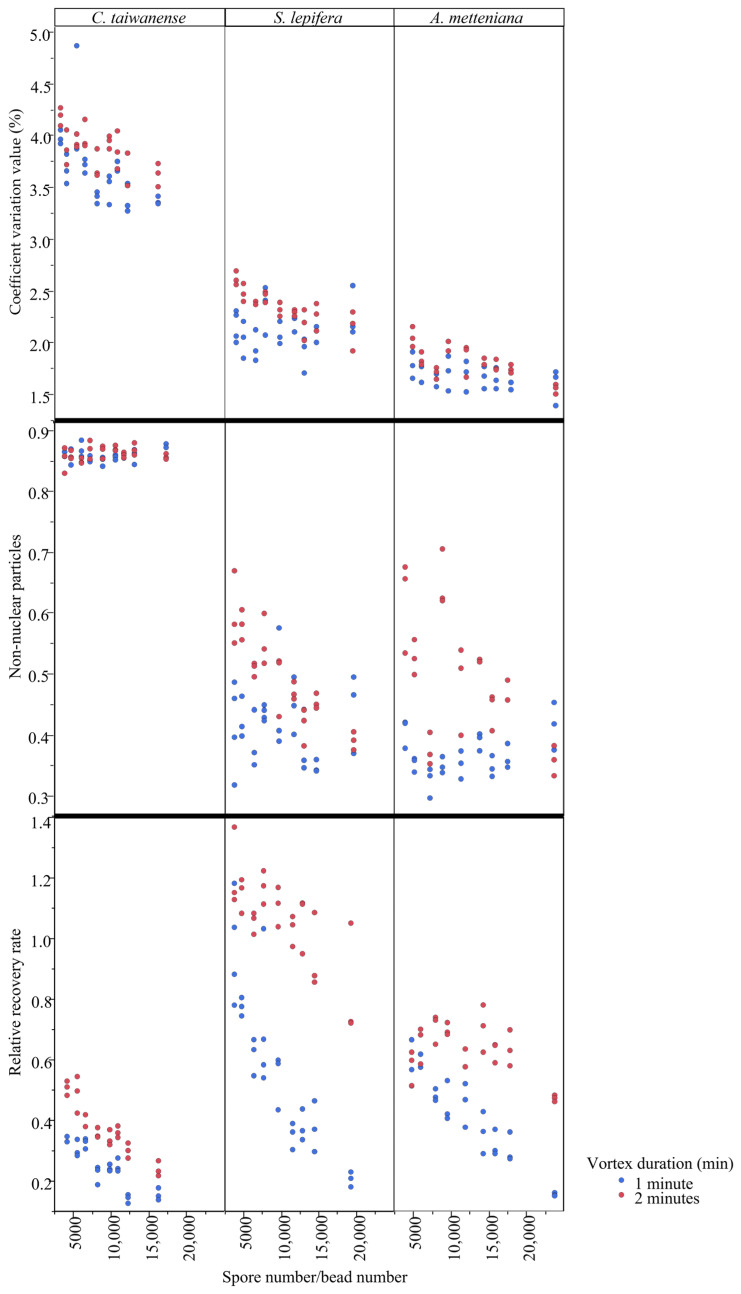
Coefficient variation (CV) value, non-nuclear particles (*np*), and relative recovery rate (*rrr*) of the three spore collections among vortex duration and spore–bead ratios.

**Figure 3 plants-12-00140-f003:**
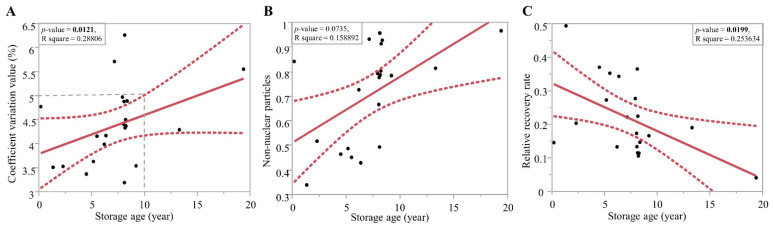
Linear regression of time and (**A**) coefficient variation value, (**B**) non-nuclear particles value, and (**C**) relative recovery rate of 4 °C-stored *Sphaeropteris lepifera* spore collections.

**Table 1 plants-12-00140-t001:** Information and spore traits of the three fern spore collections. All voucher specimens are kept in Herbarium of Taiwan Forestry Research Institute (TAIF). * Living collection of Dr. Cecilia Koo Botanic Conservation Center, no. K017423. ^#^ Living collection of National Tsing Hua University, no. TSK20180610.

Species	Family	Spore Shape	Spore Equatorial Diameter (μm)	Spore Mass (ng)	Spore Nucleus Size (μm)	Voucher No.
*Cibotium taiwanense*	Cibotiaceae	Trilete	52.67 ± 6.93	53.82 ± 8.72	3.93 ± 0.59	*Kuo4395* *
*Sphaeropteris lepifera*	Cyatheaceae	Trilete	45.27 ± 7.65	24.80 ± 5.32	4.87 ± 0.66	*Tang0001* ^#^
*Alsophila metteniana*	Cyatheaceae	Trilete	49.54 ± 6.03	36.87 ± 4.66	7.05 ± 0.65	*Lee s.n.*

**Table 2 plants-12-00140-t002:** ANOVA statistics of cytometric performance using different bead vortex conditions of the three spore collections. * *p*-value < 0.05.

Source	*Cibotium* *taiwanense* “*Kuo4395*”	*Sphaeropteris* *lepifera* “*Tang0001*”	*Alsophila* *metteniana* “*Lee s.n.*”
Coefficient variation value			
Vortex duration	**0.0326 ***	**0.0006 ***	0.419
Spore amount	**<0.0001 ***	**0.0043 ***	0.7611
Bead amount	**0.0009 ***	0.1592	0.2416
Vortex duration × spore amount	**0.0157 ***	**0.0034 ***	0.3816
Vortex duration × bead amount	**0.0172 ***	0.968	0.2134
Spore amount × bead amount	0.071	**0.0267 ***	0.9362
Vortex duration × spore amount × bead amount	0.4957	0.3742	0.8445
Non- nuclear particles			
Vortex duration	**0.0226 ***	**0.0034 ***	0.0805
Spore amount	0.4306	0.2256	**0.0073 ***
Bead amount	0.2755	0.6042	**0.0205 ***
Vortex duration × spore amount	0.2463	**0.0113 ***	**0.0017 ***
Vortex duration × bead amount	0.1704	0.4766	**0.0006 ***
Spore amount × bead amount	0.9801	**0.0307 ***	0.0746
Vortex duration × spore amount × bead amount	0.9006	0.1822	0.0838
Relative recovery rate			
Vortex duration	**<0.0001 ***	**<0.0001 ***	**<0.0001 ***
Spore amount	**<0.0001 ***	**0.0003 ***	**<0.0001 ***
Bead amount	**<0.0001 ***	**0.0008 ***	**0.0265 ***
Vortex duration × spore amount	**0.0124 ***	0.1631	0.2421
Vortex duration × bead amount	**0.0026 ***	0.3009	**0.0017 ***
Spore amount × bead amount	0.307	0.3498	0.4916
Vortex duration × spore amount × bead amount	0.1217	0.5125	**0.0215 ***

**Table 3 plants-12-00140-t003:** C-value estimates of the three spore collections using different buffers and internal standards and inferred by interpolation.

Spore Collection	Buffer	Internal Standard (s)	Mean 1C-Value [(SD) pg]	Mean CV Value [(SD)%]
*Cibotium taiwanense* “*Kuo4395*”	LB01	*Pisum sativum* cv. Ctirad	4.813 (0.010)	3.217 (0.383)
	Beckman	*P. sativum* cv. Ctirad	4.903 (0.025)	3.535 (0.127)
	GPB	*P. sativum* cv. Ctirad	5.123 (0.061)	4.258 (0.083)
	GPB	*Nicotiana tabacum* cv. Xanthi	5.144 (0.018)	3.845 (0.140)
	GPB	*Isoetes taiwanensis*	5.219 (0.135)	3.373 (0.06)
*Sphaeropteris lepifera* “*Tang0001*”	Beckman	*P. sativum* cv. Ctirad	6.751 (0.010)	2.303 (0.081)
	GPB	*P. sativum* cv. Ctirad	6.925 (0.052)	3.822 (0.285)
	LB01	*P. sativum* cv. Ctirad	6.997 (0.004)	2.213 (0.031)
	GPB	*N. tabacum* cv. Xanthi	7.239 (0.013)	4.497 (0.063)
	GPB	*I. taiwanensis*	7.717 (0.038)	2.478 (0.061)
*Alsophila metteniana* “*Lee s.n.*”	GPB	*P. sativum* cv. Ctirad	18.32 (0.062)	2.86 (0.062)
	LB01	*N. tabacum* cv. Xanthi	18.883 (0.057)	3.99 (0.185)
	Beckman	*N. tabacum* cv. Xanthi	19.199 (0.047)	2.35 (0.079)
	GPB	*N. tabacum* cv. Xanthi and *V. faba* cv. Inovec	19.289 (0.032)	1.923 (0.42)
	GPB	*V. faba* cv. Inovec	19.396 (0.006)	2.235 (0.022)
	GPB	*N. tabacum* cv. Xanthi	20.324 (0.031)	2.578 (0.125)

**Table 4 plants-12-00140-t004:** C-value estimation of the three spore collections using different buffers and internal standards by interpolation.

Spore Collection	Buffer	Internal Standard (s)	Mean 1C-Value [(SD) pg]	Mean CV Value [(SD)%]
*Cibotium taiwanense* “*Kuo4395*”	GPB	*Secale cereale* cv. Dankovske	3.536 (0.038)	3.278 (0.279)
*Sphaeropteris lepifera* “*Tang0001*”	GPB	*Nicotiana tabacum* cv. Xanthi and *Vicia. faba* cv. Inovec	6.849 (0.022)	3.080 (0.138)
*Alsophila metteniana* “*Lee s.n.*”	GPB	*V. faba* cv. Inovec	16.595 (0.028)	2.207 (0.07)
	GPB	*V. faba* cv. Inovec (*2n* nuclei) and *V. faba* cv. Inovec (*4n* nuclei)	16.786 (0.075)	2.207 (0.07)

**Table 5 plants-12-00140-t005:** Recommended C-values of the three spore collections inferred by linear regression.

Spore Collection	Internal Standards in Regression Samples	1C-Value (pg)	1C-Value (Gbp) *
*Cibotium taiwanense* “*Kuo4395*”	*Nicotiana tabacum* cv. Xanthi, *Pisum sativum* cv. Ctirad, *Isoetes taiwanensis*	5.058	4.947
*Sphaeropteris lepifera* “*Tang0001*”	*N. tabacum* cv. Xanthi, *P. sativum* cv. Ctirad, *I. taiwanensis*	7.117	6.960
*Alsophila metteniana* “*Lee s.n.*”	*N. tabacum* cv. Xanthi, *P. sativum* cv. Ctirad, *Vicia. faba* cv. Inovec	19.379	18.953

* based on 1 pg = 0.978 Gbp.

## Data Availability

All the data are available at the Appendix A of this article.

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
