# Peer review of "Fern Spores—“Ready-to-Use” Standards for Plant Genome Size Estimation Using a Flow Cytometric Approach"

_plants, 2022, doi:10.3390/plants12010140_

Round 1

Reviewer 1 Report

This study extends the utility of flow cytometry for the measurement of spore genome sizes, using a method initially developed by Kuo et al. (2017), by comparing leaf and spore material stored in various conditions/durations, using light microscopy and flow cytometry. The most impactful aspects of this study are, (1) the careful optimization of methods for extracting genome size data for focal species (Cibotium, Alsophila, and Sphaeropteris—all tree ferns), (2) the establishment of living and frozen spore collections to serve as global genome size standards, and (3) the novel evidence that spores collected from old herbarium material do not provide accurate estimates of genome size by FCM. Overall, this is a useful study that offers timely methodological improvements for analyses of genome size in spore-bearing plants.

Despite many strengths of this study, some critical suggestions did arise upon review. Firstly, I am curious about the finding that spore and leaf material differed in genome size for a given sample. Can you elaborate on this finding or provide thoughts on cause(s) or implications? (Something beyond these points, perhaps? “…were likely caused by tissue-specific effects…”  or “previous studies suggest that 1C-value estimates using leaf materials are usually lower than that using spores, and seemingly underestimated [18,20,21,23,24]”?) Secondly, given that this study proposes specific living (and frozen) collections to serve as global genome size standards, I encourage the authors to consider reporting chromosome counts for source plants. In addition to these points, I have provided detailed suggestions below and as tracked changes in the accompanying PDF, where I have also included minor grammatical/phrasing recommendations.

Abstract

“…were examined for their nucleus quality and quantity…” I’m not sure I understand what you mean by this, please revise.

P.1 Introduction

“…the most efficient and cost-effective method…”  Arguably, but fuelgen is efficient and cost-effective also. Is flow cytometry more accurate?

P.2.

“…most ferns can reproduce by selfing...”  Most studies show, perhaps surprisingly, that outcrossing predominates among homosporous ferns—despite having bisexual gametophytes. It is true that producing bisexual gametophytes allows for extreme inbreeding that leads to completely homozygous offspring (e.g., when n = 1x), but this is not the same as “most ferns can reproduce by selfing”; please revise this statement.

In the sentence that begins, “We first examined spore traits…”, how do nuclear size and genome size differ?

Do you know the approximate room temperature? If not, please describe the general conditions (e.g., air conditioned or not, humid or dry, etc.)

The phasing of this sentence is awkward; please revise. “Compared to diploids, polyploids due to their enlarged genome size might cost double or more to complete their whole genome sequencing, and are thereby undesirable for such projects.”

What is meant by the following statement? “Some spore traits need to be investigated in advance, and treatments shall be assessed for maintaining the quality and quantity of nuclei during extraction.“ Specifically, which spore traits need to be investigated and why?

Note typo / phrasing error: “Last, storage age is one of important factors..”

Poor/incorrect phrasing, please revise: “To develop spores as new C-value standards for plants, we chose spores of ferns as materials because high quantity of spores per individual is highly accessible in these plants.”

In this sentence, please specify what you mean by “size”: “To develop spores as new C-value standards for plants, we chose spores of ferns as materials because high quantity of spores per individual is highly accessible in these plants.”

Upon first mention of formal taxon names at or below the genus level, please indicate authority name

Please clarify, whose extraction protocol? “in order to optimize their extraction protocol of spore nuclei, we tested different bead-vortex conditions…”

Fig. 1. Consider elaborating within the figure legend here.

2.1

The following sentence is unclear, please revise: “Despite these similarities, their intracellular contents tend to be different in proportion.”

Given that these samples are being proposed as critical standards, I recommend depositing several duplicates into herbarium collections worldwide.

Methods and Materials

4.1

“However, because the triploid apomictic individual of D. varia yielded insufficient spores, we were unable to complete full assessment for this species. Three spore collections for the three remaining candidates are detailed in Table 1.” Did you or did you not include D. varia in any of your analyses? If yes, please include source details for this taxon in Table 1.

“…selected candidates after preliminary tests.” Please elaborate on what you mean by “preliminary tests”

Regarding “spore sources” (1st paragraph 4.1), were any of these herbarium specimens? If so, please highlight this here, as well as relevant herbarium codes.

4.2

“..with fixed magnification…” doesn’t make sense here, maybe you mean “fixed amount”?

“..counted the concentration of spore[s]…” Please elaborate, what was your estimate of concentration? Did you count a certain number for a certain area on the slide?

“…sizes of 30 spore nuclei were measured…” What was your metric for measuring size?

Author Response

Despite many strengths of this study, some critical suggestions did arise upon review. Firstly, I am curious about the finding that spore and leaf material differed in genome size for a given sample. Can you elaborate on this finding or provide thoughts on cause(s) or implications? (Something beyond these points, perhaps? “…were likely caused by tissue-specific effects…”  or “previous studies suggest that 1C-value estimates using leaf materials are usually lower than that using spores, and seemingly underestimated [18,20,21,23,24]”?)

Re: we appreciate the reviewer raising this point. We provided more details to explain the possible causes. “In particular, chromatin is found increasingly decondensed in plant nuclei during the transition from a vegetative fate into a reproductive one, for example sporogenesis of spore mother cells [40], and thus become much accessible to DNA staining chemicals.”

Secondly, given that this study proposes specific living (and frozen) collections to serve as global genome size standards, I encourage the authors to consider reporting chromosome counts for source plants. 

Re: we appreciate the reviewer raising this point. Unfortunately, we are unable to timely provide chromosome counts for these source plants. Nonetheless, we believe their ploidies are same as previous reports, which also have ploidy information. In addition, the genome size data of these three collections match the expectations from congeneric or conspecific individuals with ploidy inference. We detailed these in the revised Materials and Methods.

“These three collections were revealed with the genome sizes (Figure 1) matching that from previous conspecific or congeneric reports with ploidy inferences [42,43]. As a result, C. taiwanense “Kuo4395”, S. lepifera “Tang0001”, and A. metteniana “Lee s.n.” are assumed to be respectively a diploid, diploid, and tetraploid. These ploidies are also identical with previous cytological reports of the same species known from the same source region (i.e. Taiwan) [59]."

In addition to these points, I have provided detailed suggestions below and as tracked changes in the accompanying PDF, where I have also included minor grammatical/phrasing recommendations.

Re:  thank you for these! We revised accordingly.

Abstract

“…were examined for their nucleus quality and quantity…” I’m not sure I understand what you mean by this, please revise.

Re: we revised this part with further clarification. “…their spore nuclei were extracted, and the quality and quantity of these nucleus extractions through storage ages were examined.”

P.1 Introduction

“…the most efficient and cost-effective method…”  Arguably, but fuelgen is efficient and cost-effective also. Is flow cytometry more accurate?

Re: we clarified this part with more details. “By far, flow cytometry (FCM) proves the most efficient and cost-effective approach to infer accurate genome size in plants [7,8].”

P.2.

“…most ferns can reproduce by selfing...”  Most studies show, perhaps surprisingly, that outcrossing predominates among homosporous ferns—despite having bisexual gametophytes. It is true that producing bisexual gametophytes allows for extreme inbreeding that leads to completely homozygous offspring (e.g., when n = 1x), but this is not the same as “most ferns can reproduce by selfing”; please revise this statement.

Re: we revised this part with further clarification. “Last, selfing lines can be generated in most ferns.”

In the sentence that begins, “We first examined spore traits…”, how do nuclear size and genome size differ?

Re: we clarified this part with more details. “We first examined spore traits, including their diameter, mass, nucleus size, and genome size in different fern species,..”

Do you know the approximate room temperature? If not, please describe the general conditions (e.g., air conditioned or not, humid or dry, etc.)

Re: we added such details in the revised manuscript.

The phasing of this sentence is awkward; please revise. “Compared to diploids, polyploids due to their enlarged genome size might cost double or more to complete their whole genome sequencing, and are thereby undesirable for such projects.”

Re: we revised this part with further clarification. “Compared to diploids, polyploids, due to their enlarged genome size, might cost double (or more) to complete whole genome sequencing, and are thereby undesirable for such projects.”

What is meant by the following statement? “Some spore traits need to be investigated in advance, and treatments shall be assessed for maintaining the quality and quantity of nuclei during extraction.“ Specifically, which spore traits need to be investigated and why?

Re: we deleted this part to avoid confusion.

Note typo / phrasing error: “Last, storage age is one of important factors..”

Re: revised accordingly.

Poor/incorrect phrasing, please revise: “To develop spores as new C-value standards for plants, we chose spores of ferns as materials because high quantity of spores per individual is highly accessible in these plants.”

In this sentence, please specify what you mean by “size”: “To develop spores as new C-value standards for plants, we chose spores of ferns as materials because high quantity of spores per individual is highly accessible in these plants.”

Re: we revised this part with further clarification. “To develop spores as new C-value standards for plants, we chose ferns for our study because a high quantity of spores per individual is easily accessible in these plants [26]⁠. We first examined spore traits, including their diameter, mass, nucleus size, and genome size in different fern species, and selected three collections.”

Upon first mention of formal taxon names at or below the genus level, please indicate authority name

Re: revised accordingly.

Please clarify, whose extraction protocol? “in order to optimize their extraction protocol of spore nuclei, we tested different bead-vortex conditions…”

Re: we clarified in the revised manuscript. “Further, in order to optimize extraction protocol of their spore nuclei, we tested different bead-vortex conditions, and determined the best acquisition with either a considerable quality or quantity for spore nuclei among them.”

Fig. 1. Consider elaborating within the figure legend here.

Re: we elaborated these in the revised manuscript.

2.1

The following sentence is unclear, please revise: “Despite these similarities, their intracellular contents tend to be different in proportion.”

Re: we clarified this part with more details. “The nucleus sizes positively correlate with their genome sizes, and both change in the same order of A. metteniana > S. lepifera > C. taiwanense (Table 1). however, the changes of spore size (i.e., equatorial diameter) and mass are shown with a different trend that does not correspond to their nucleus or genome size. Specifically, their intracellular space occupied by a nucleus tends to be different in proportion.”

Given that these samples are being proposed as critical standards, I recommend depositing several duplicates into herbarium collections worldwide.

Re: their specimens, including duplicates, were already in TAIF herbarium. Nonetheless, we are happy to collect new specimens from these living source plants in near future, and send to other herbaria.

Methods and Materials

4.1

 “However, because the triploid apomictic individual of D. varia yielded insufficient spores, we were unable to complete full assessment for this species. Three spore collections for the three remaining candidates are detailed in Table 1.” Did you or did you not include D. varia in any of your analyses? If yes, please include source details for this taxon in Table 1.

Re: unfortunately, we not yet have any other data, expecting for a preliminary survey of its genome size (as cited in Materials and Methods).

“…selected candidates after preliminary tests.” Please elaborate on what you mean by “preliminary tests”

Re: we elaborated this with more details. “…selected candidates after preliminary tests by quickly checking their flow cytometric performance.”

Regarding “spore sources” (1st paragraph 4.1), were any of these herbarium specimens? If so, please highlight this here, as well as relevant herbarium codes.

Re: we added all these details in the revised Table 1.

4.2

“..with fixed magnification…” doesn’t make sense here, maybe you mean “fixed amount”?

 “..counted the concentration of spore[s]…” Please elaborate, what was your estimate of concentration? Did you count a certain number for a certain area on the slide?

Re: This part appears to be clarified as in our formula listed there. The concentration means particles/volume so it is not based on area on a slide.

“…sizes of 30 spore nuclei were measured…” What was your metric for measuring size?

Re: we clarified this part as “For each species, 30 spore nuclei were measured by Image Pro plus 5.0.0 (Media Cybernetics, Silver Spring, USA).”

Reviewer 2 Report

I don't want to be too harsh, but the article is built on the premise that fern spores can serve as an FCM standard. For such a purpose, however, the author's choice of ferns is beyond my understanding of how it could relate in any way to FCM standards. First of all, all three ferns are restricted to SE Asia or, in the case of Cibotium taiwanense, are even endemic to Taiwan. This is in stark contrast to the recommended availability of standards for general use. How can these ferns propagate outside their home country? Or are the authors assuming export of spores for use as an FCM standard? I really don't understand.

The second significant flaw in this study (and unfortunately quite fundamental) is the lack of data on genome size stability across the entire range of distribution. There are only data for Sphaeropteris lepifera (Supplementary Table 1) that indicate the absolute uselessness of this species as a possible candidate for an FCM standard, due to intraspecific variation. A simple comparison of the 1C-values listed in the table provides a minimum-maximum range of 6.90-8.34, representing 21% of the variability, curiously recorded in the Taipei city area. This contradicts any meaningful use of this species as an FCM standard.

Frankly, I cannot recommend the article in its current form because the data itself is in direct conflict with the proposed use. Alternatively, the data could be used as an extension of the already known approach to estimating genome size from spores, but to do so the paper would need to be completely rewritten.

Author Response

We appreciate the review raising this concern. However, he or she probably misunderstood how we decided the candidate collections. We hope that the following explanation can alleviate this concern from the reviewer.

In fact, we referred the three standards as three source plants rather than three species. To better clarify these, we add the collection numbers of these source plants behind their species names in the revised manuscript. Particularly, as suggested in previous flow cytometric studies, using the same breeding lines from the established standards is critical to avoid infraspecific variation of genome size from the standards, and to accurately infer genome size of a sample. As stated in our manuscript, all our institutions are willing to share the recourses of these spore materials for any needs in future research. Therefore, all these collections, even being endemic to Taiwan, are no problem to be distributed oversea. Even for those well-established standards of crops, many researchers still need to request their seeds from labs in a different country, similarly as in our current study (see in our acknowledgement). We believe such share of biological materials is a general rule for life science research. It is not reasonable to discriminate research materials whether they are well known or novel but still poorly known ones. In fact, dry fern spores can be easily distributed and mailed as done long-term in several institutions:

https://www.amerfernsoc.org/spore-exchange-background

https://ebps.org.uk/ferns/growing/spore-exchange/

Or even been commercialized:

https://www.carolina.com/c-fern/c-fern-spores-wild-type/FAM_156728.pr

Based on these, we don’t think this concern from the reviewer is a critical point for our study.

Reviewer 3 Report

Dear Authors, 

the results presented in the manuscript are very interesting. 

However, it is misunderstood that manuscript entitled: "Fern Spores-“Ready-to-Use” Standards for Plant Genome Size Estimation" which is related to use of the flow cytometry to optimizing protocol to genome size estimation, you present only the recalculated results. Recalculations in the tables are clear. However, it is unclear why you compare the non-nuclear particles with the nuclear ones. There are no clear description of what is that. It is important for the scientists who are not specialist in the subject. 

I am the cytologist, and I know that presentation of the original charts obtained during the FCM analyses is the basis, especially that you optimized the protocol. The charts would make this paper more attractive, even if they are not so blank.  

Moreover, the description in the headline of the tables are insufficient. There should be added more information abut the content. 

Best regards

Author Response

“However, it is misunderstood that manuscript entitled: "Fern Spores-“Ready-to-Use” Standards for Plant Genome Size Estimation" which is related to use of the flow cytometry to optimizing protocol to genome size estimation, you present only the recalculated results. Recalculations in the tables are clear.”

Re: we modified the title into “Fern spores- “ready-to-use” standards for plant genome size estimation using a flow cytometric approach” to much fit the aims of our study.

“However, it is unclear why you compare the non-nuclear particles with the nuclear ones. There are no clear description of what is that. It is important for the scientists who are not specialist in the subject.”

Re: we added more explanations for this in the materials & methods section. “A lower CV value represents higher quality of nuclei, while, greater np implies a higher proportion of non-nuclear particles, which likely resulted from over-disruption during bead vortexing and caused noisy signals for flow cytometric inference.”

“I am the cytologist, and I know that presentation of the original charts obtained during the FCM analyses is the basis, especially that you optimized the protocol. The charts would make this paper more attractive, even if they are not so blank.” 

Re: we added a figure (i.e. the new Figure 1) with charts obtained during the FCM analyses.

“Moreover, the description in the headline of the tables are insufficient. There should be added more information abut the content.”

Re: we added more relevant details in these table titles.

Round 2

Reviewer 3 Report

Dear Authors, 

I accept all corrections. 

Author Response

Thank you!